# Implementation of antimicrobial stewardship programs: A study of prescribers' perspective of facilitators and barriers

Emelda E. Chukwu[1]*, Dennis Abuh[1], Ifeoma E. Idigbe[1], Kazeem A. Osuolale[1], Vivian Chuka-Ebene[1,2], Oluwatoyin Awoderu[1], Rosemary A. Audu[1], Folasade T. Ogunsola[1,3]

1 Antimicrobial Resistance Research group, Nigerian Institute of Medical Research, Yaba, Lagos State, Nigeria, 2 Pharmacy Department, Lagos University Teaching Hospital, Lagos, Lagos State, Nigeria, 3 Department of Medical Microbiology and Parasitology, College of Medicine, University of Lagos, Lagos, Nigeria

* emeldachukwu123@gmail.com

**Data Availability Statement:** All relevant data are within the manuscript and its Supporting Information files

## Abstract

### Background

Despite promising signs of the benefits associated with Antimicrobial Stewardship Programs (ASPs), there remains limited knowledge on how to implement ASPs in peculiar settings for a more elaborate impact. This study explored prescriber experiences and perceptions of the usefulness, and feasibility of strategies employed for the implementation of antimicrobial stewardship (AMS) interventions as well as challenges encountered.

### Methods

This is a cross-sectional mixed-method survey of prescribers' perspective of the facilitators and barriers of implementing ASP. The quantitative approach comprised of a semi-structured questionnaire and data collected were analyzed using SPSS version 26 while the qualitative approach used focus group discussions followed by content analysis.

### Results

Out of the thirty people that participated in the workshop, twenty-five completed the questionnaires which were analyzed. The respondents included 15 (60.0%) medical doctors and 10 (40.0%) pharmacists. The mean age of the respondents was 36.39±7.23 years with mean year of practice of 9.48±6.01 years. Majority of them (84.0%) were in a position to provide input on the implementation of AMS in their facilities, although their managements had the final decision. The pharmacists (100%) were more likely to agree that antibiotic resistance was a problem for their practice than the medical doctors (78.6%) while equal number (80.0%) of respondents (pharmacists and medical doctors) believed that inappropriate prescribing was a problem. *Having a specialized and dedicated team with effective monitoring* was recognized as crucial for effective ASP while *inadequate personnel* was identified as a major barrier. We identified *stakeholder's engagement*, *policies and regulation*, as well as *education* as themes for improving AMS in the country.

**Funding:** he author(s) received no specific funding for this work

**Competing interests:** The authors have declared that no competing interests exist.

**Abbreviations:** AMR, Antimicrobial resistance; AMS, Antimicrobial stewardship; AMU, Antimicrobial use; ASP, Antimicrobial stewardship program; CDC, Center for diseases control and prevention; COM-B, capability, opportunity, motivation and behavior; FGD, Focus Group Discussion; GLASS, Global antimicrobial resistance and use surveillance system; LMIC, Low- or Middle —Income Countries; NAP, National action plan; WHO, World Health Organization.

## Conclusion

The results gave insight into the prescribers' perspective on the facilitators and barriers to antimicrobial stewardship; challenges and possible solutions to implementing ASPs in health facilities in Lagos State. We further identified pertinent contextual factors that need to be addressed when developing ASPs in healthcare facilities in a resource-poor setting.

## Introduction

Antibiotic resistant bacteria cause infections that are difficult to treat and characteristically associated with increased morbidity, mortality, longer hospital stays, and excess health care costs [1–3]. Antibiotic use is a major driving force that selects for antibiotic resistant bacteria and approximately 50% of antibiotics used in hospitals have been estimated to be inappropriate [4]. The increasing global threat of antimicrobial resistance (AMR) has brought to the fore, the need for interventions to contain the emergence and transmission of AMR genes among bacteria pathogens. Antimicrobial stewardship programs (ASPs) have proven to be efficient in the short term, with no clear evidence of what the successful components are for a sustainable change in prescribing practices [5]. A systematic review of antimicrobial prescribing studies in hospitals suggests that sustainability of ASPs may be improved with a better understanding of behavioral determinants of prescribing [6]. Another review concluded that cultural, contextual and behavioral factors need to be addressed to influence antimicrobial use [7].

Antibiotic stewardship interventions are crucial to slow the development of resistance and it involves coordinated strategies designed to improve the appropriate use of antibiotics, by promoting the optimal drug, dose, duration, and route of administration [4]. Effective ASP will require multidisciplinary approach and engagement towards providing quality health care and ensuring patient safety. Despite the progress made on antibiotic stewardship in high income countries, most healthcare facilities especially in low- and middle-income countries still face barriers, such as limited financial and staff resources, in the implementation and maintenance of successful comprehensive antibiotic stewardship programs [8].

The global antimicrobial resistance and use surveillance system (GLASS) identified antimicrobial use (AMU) as a major driver of AMR and seeks to monitor AMU by collating the nationally aggregated data on antimicrobial consumption, as well as promoting studies on antibiotic prescription practices [9]. Nonetheless, generation of quality data that can inform national policy has continually proven to be a major challenge in most low- or middle-income countries (LMICs). Part of the key strategy of the Nigerian National Action Plan (NAP) to tackle AMR is to promote rational access to antibiotics and antimicrobial stewardship through optimizing of antimicrobial prescribing and dispensing in human and animals [10]. However, the extent to which the implementation of this 5-year strategic plan has been achieved remains to be ascertained. A recent evaluation of NAP implementation in WHO Africa region reveals the need for a context-driven approach, taking into account the specificities and peculiarities inherent in the respective member states [11].

The Center for Diseases Control and Prevention [CDC] has outlined core elements to improve antibiotic use in hospitals and nursing homes settings [12, 13]. Previous cross-sectional assessment revealed minimal ASP activities across healthcare facilities in Lagos State with sub-optimal performance in ASP implementing facilities [14]. However, it is not well understood which factors promote or hinder the implementation and maintenance of these programs within the Nigerian context. Both qualitative and quantitative studies have been carried out on this topic [15–17], however, we explored the use of mixed-method to investigate

facilitators and barriers to implementing AMS interventions as well as factors influencing anti-microbial prescribing practices among hospital doctors and pharmacists. This combined approach allows for the collection of numerical and measurable data while at the same time exploring the perspectives of the participants with a view to explain, validate or complement quantitative data. The objective of this study is to identify barriers and facilitators to antibiotic stewardship within selected (primary, secondary, tertiary and private/faith-based) healthcare facilities in Lagos State. This was achieved through qualitative interviews with medical doctors and pharmacists saddled with the responsibilities of making antibiotic recommendations within their facilities. This study aimed to identify facilitators, barriers and strategies in imple-menting AMS intervention in healthcare facilities in Lagos State and explored the potential role(s) of doctors and pharmacists in this intervention.

## Methodology

### Study setting

The study was carried out during a 3-day training workshop on Antimicrobial Stewardship for prescribers held at the Nigerian Institute of Medical Research (NIMR) from 15th to 17th November 2021. The workshop was titled "Improving antibiotic-related patient safety through proper antibiotic prescribing". Facilities were selected to reflect the different categories of healthcare facilities in Lagos State i.e., the tertiary, secondary, primary and private/faith-based healthcare facilities, taking into account the number of different levels of healthcare facilities as classified by the National Strategic Health Development Plan [18] and identified in our previ-ous needs assessment [14]. To ensure management buy-in and engagement in future imple-mentation plans, the management of the selected facilities were asked to nominate key players in their AMS programs to participate in the training. The participants were medical doctors and pharmacists from selected healthcare facilities across Lagos State who have important roles in improving antibiotic use.

### Study design and sampling technique

A cross-sectional descriptive mixed method study to investigate the prescriber's perspective on the facilitators, barriers and strategies for implementation of ASP in healthcare facilities in Lagos State. The research design comprised both quantitative and qualitative methodologies. Quantitative data was collected through a semi-structured questionnaire to gather demo-graphic information and assess the current level of ASP implementation in individual health-care facilities. The questionnaire and interview guide for this study were developed by the study team from review of similar studies [19, 20], and can be found in the supplementary material (S1 Questionnaire and S1 File). The second phase involved a qualitative approach using focus group discussions followed by conventional content analysis. Purposeful sampling was used to recruit prescribers (doctors and pharmacists) attending the AMS training. All par-ticipants who attended the 3-day training on AMS and willingly provided their written consent were included in the questionnaire-based survey and/or focus group discussion. Qualitative data was collected using a semi-structured interview guide.

### Focus Group Discussions (FGD)

FGD was conducted to obtain the views of medical doctors and pharmacists regarding the cul-ture of antibiotic use at their healthcare facilities and on barriers and facilitators to antibiotic stewardship. Culture of antibiotic use was defined as the values, attitudes, and practices sur-rounding antibiotic use shared by the medical staff. Culture represents the way antibiotics are

used and the attitudes of staff about how antibiotics are used. The interview questions and the analysis were guided by the framework used by Ploeg et al [21] to explore and understand individual (e.g., a clinician's knowledge and attitudes), organizational (e.g., leadership support, teamwork), and environmental (e.g., support from national professional body) factors influencing inappropriate antibiotic use and antibiotic stewardship. Two sets of FGDs were conducted with 10 participants in each set. One set was conducted with medical doctors and another set was conducted with pharmacists. The FGDs had proper representation of participants based on demographics (Gender, age,) type of practice (public, private), range of experiences (early career, mid-career and advanced career clinicians and pharmacists) and the numbers of specialist in each group providing unique and vast knowledge.

## Data analysis

Quantitative data was summarized using descriptive statistics and a significance level of P < 0.05 was considered statistically significant. The analysis of quantitative data was performed using SPSS version 26.0. For the qualitative approach, data were collected using Focus Group Discussion with consenting participants who attended the training on AMS.

All interviews were audio recorded and transcribed verbatim. Barrier and facilitator themes were identified from the transcripts and mapped using the COM-B (capability, opportunity, motivation and behavior) model [22]. Similarly, themes were identified from the transcripts regarding the roles of prescribers in AMS intervention. Employing the inductive approach, data were transcribed and coded using open coding to develop a codebook. The codebook included a list of question prompts, initial codes, and code meanings. The team developed a coding tree, deriving themes until reaching thematic saturation. Codes were reviewed, compared, analyzed, and sorted into categories to reflect consistent and overarching themes as described by Hsieh and Shannon [23].

## Ethical clearance

Ethical clearance for this study was obtained from the Nigerian Institute of Medical Research Institutional Review Board (IRB/21/083). Participants were provided detailed written consent forms about the study and were informed of their rights to participate or decline in the study. Those who agreed to participate, signed a written consent form, and partook in the study. Information obtained from respondents were treated as confidential with no identifier.

## Results

### Quantitative study

Out of the thirty individuals who participated in the workshop, 26 (86.7%) consented and were included in the quantitative study. Twenty-six questionnaires were received but one was excluded due to incomplete filling (less than 50%). Therefore, a total of 25 questionnaires were analyzed with 15 (60%) being from medical doctors and 10 (40.0%) from pharmacists (Table 1).

The participants' mean age was 36.39±7.23 years with mean year of practice of 9.48±6.01 years. Among the respondents, the majority (84.0%) held positions that allowed them to provide input on the implementation of AMS in their respective healthcare facilities. However, the final decisions were made by their facilities' management while one participant was a primary decision maker (Fig 1). About a half of the participants either did not have or were not sure if there existed an antimicrobial stewardship team in their facility while 28% were members of their facilities' stewardship team (Table 1).

**Table 1. Demographics and practice characteristics of participants for quantitative study.**

| Variable | Number (%) |
|---|---|
| **Gender (N = 25)** | |
| Female | 14 (56.0) |
| Male | 11 (44.0) |
| **Profession (N = 25)** | |
| Medical doctor | 15 (60.0) |
| Pharmacist | 10 (40) |
| **Age groups[a] (N = 23)** | |
| 25–29 | 5 (20.0) |
| 30–34 | 5 (20.0) |
| 35–39 | 5 (20.0) |
| 40–44 | 5 (20.0) |
| 45 and above | 3 (8.0) |
| **Year of practice N = 25** | |
| 1–5 | 9 (36.0) |
| 6–10 | 4 (16.0) |
| 11–15 | 9 (36.0) |
| 16 and above | 3 (12.0) |
| **Healthcare category** | |
| Primary healthcare centres | 3 (12.0) |
| Secondary Healthcare (General hospitals) | 9 (36.0) |
| Tertiary healthcare (Teaching hospitals) | 7 (28.0) |
| Private/ Faith-based hospitals | 6 (24.0) |
| **Does your facility have an antimicrobial stewardship team?[b]** | |
| Yes | 12 (48.0) |
| No | 10 (40.0) |
| I don't know | 2 (8.0) |
| **Are you a member of your facility stewardship team?[c]** | |
| Yes | 7 (28.0) |
| No | 17 (68.0) |

Key: a = missing 2, b = missing 1, c = missing 1

The participants gave varied responses to the question on the acceptability of stewardship approaches/ interventions (Fig 2). Over half (68.0%) of the respondents agreed that ASPs were needed in healthcare settings to effectively deal with antibiotic resistance, although a number of them (16.0%) were of the opinion that practice-based reporting would be too burdensome and 68.0% indicated that they would need a lot of help to implement antibiotics stewardship interventions. Majority (72.0%) of the respondents were confident that they prescribed antibiotics more appropriately than their peers while 36.0% believed that tracking the appropriate use of antibiotics would be difficult to do in an accurate and fair manner. The respondents (88.0%) reckoned that antibiotics stewardship efforts implemented by providers would be ineffective unless also paired with efforts aimed at educating patients/parents about antibiotic resistance and antibiotic use (Fig 2).

Fig 3 depicted the responses of participants to questions on the perception of stewardship interventions. Although majority (96.0%) of the respondents believed that antibiotic resistance was a problem in Nigeria, only 84.0% agreed/strongly agreed that antibiotic resistance was a problem for their practice. All respondents agreed that inappropriate antibiotic prescribing in

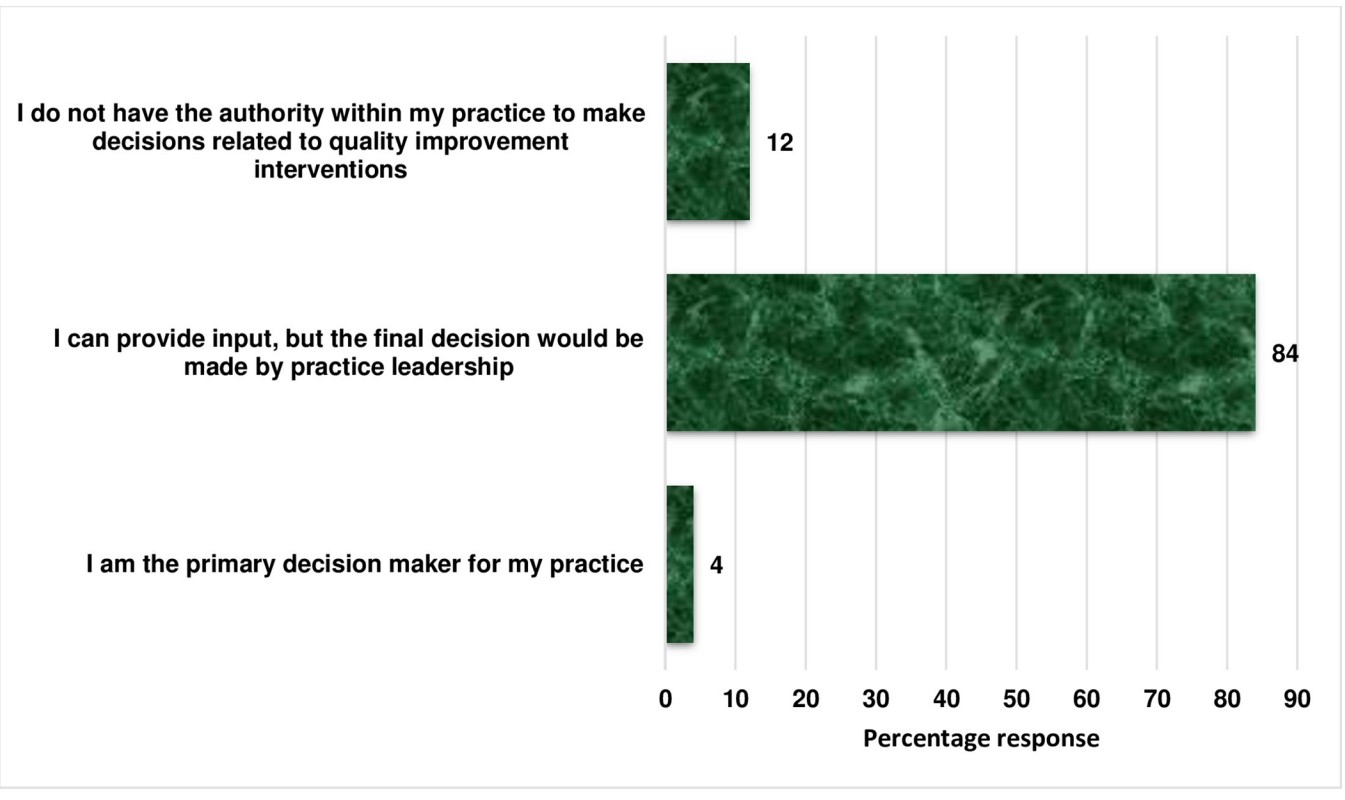

**Fig 1. Response to the question on the level of authority to implement AMS in their facilities.**

the outpatient healthcare setting accelerates the emergence of antibiotic resistance. However, 68.0% of the respondents held that the problem was not limited to outpatient healthcare setting (Fig 3).

In general, 60.0% and 80.0% of the respondents believed that inappropriate antibiotic dispensing and prescribing was a problem in their practice respectively. However, when disaggregated between medical doctors and pharmacists, equal number (80.0%) of respondents (pharmacists and medical doctors) believed that inappropriate prescribing was a problem while pharmacists were less willing (40.0%) to agree that inappropriate dispensing was a problem for their practice (Fig 4). Nevertheless, the pharmacists (100%) were more likely to agree that antibiotic resistance is a problem for their practice than the medical doctors (78.6%).

## Qualitative study results/findings

A total of 20 respondents (Clinicians = 10; Pharmacists = 10) gave their consent and participated in the focus group discussion and were between the ages of 25 to 47 years. Two FGDs were conducted consisting of 10 participants each. Of these respondents, 12 were from government facilities and 8 were from private facilities. All participants resided and worked in Lagos State. Approximately, 70% of the respondents were females with average age of 36 years.

The respondents described the level of ASPs in their facilities which ranged from early-stage consultations to advanced level of ASP. Majority of the facilities were at the early stage of development with many of the facilities possessing a mini constituted team who were entrusted with the task of monitoring and controlling the use of antibiotics in patient management.

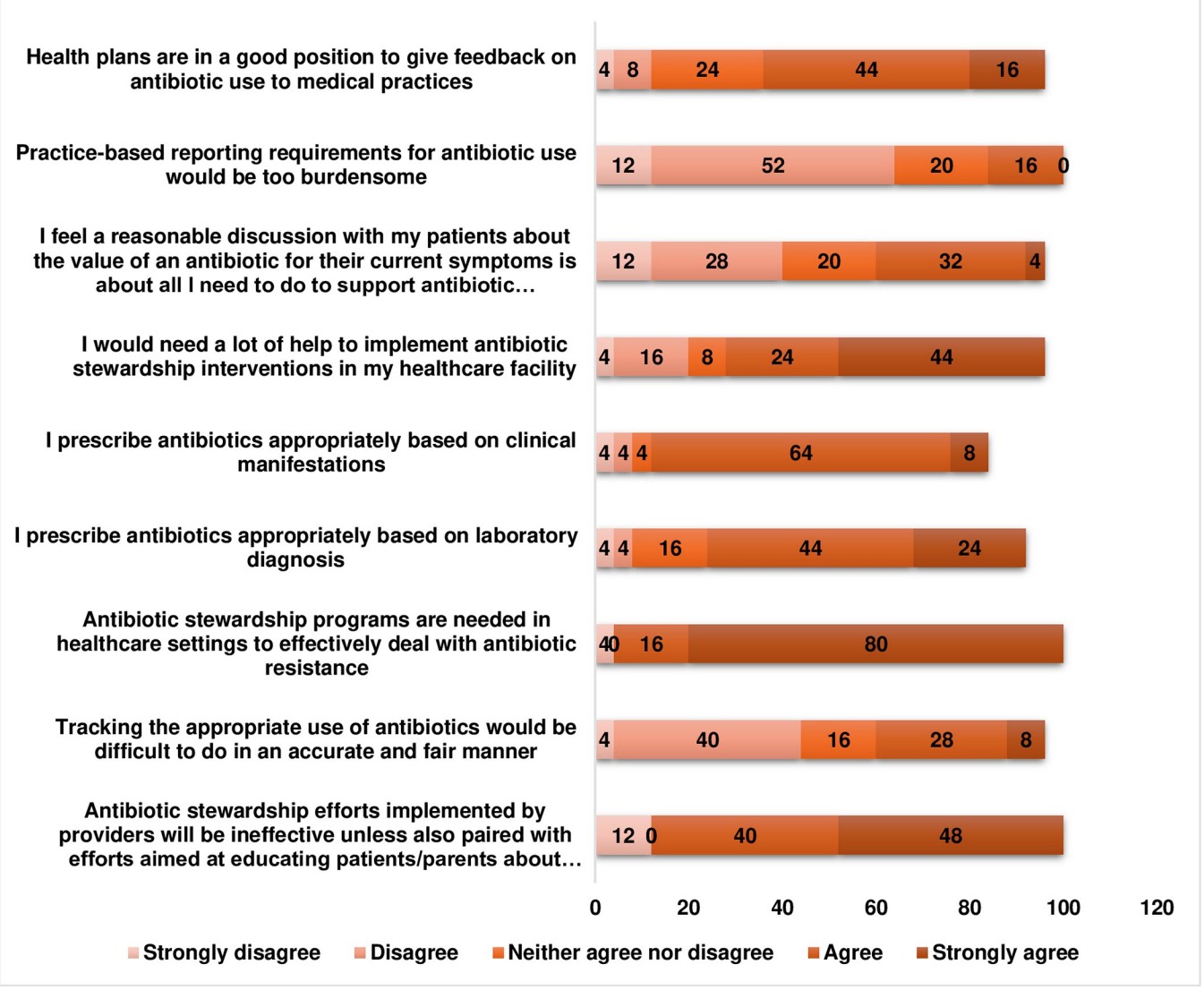

**Fig 2. Responses to questions on the acceptability of stewardship approaches/ interventions.**

"*We don't have an established antibiotics stewardship program. Yes, we don't have but I know we had infection control committee at one time, that I was a member. I think we were about two pharmacists there. So. . .then, we were doing things to reduce the overall infection rate in the hospital. We were looking at the nosocomial infection rate. So, we are doing something on the side but just not as a committee*" #Pharmacist 1

"*The antimicrobial stewardship is a small committee comprising of our medical directors, the health service manager, and pharmacists who work together to. . . you know. . . streamline the supply of drugs coming in and going out. But on a daily basis, the antimicrobial stewardship is headed by a consultant. We are trying to control the use of antibiotics amongst the medical team.*" #Clinician 1

"*The department that does antimicrobial monitoring. That's what we call it. And we do it basically for one patient or mainly for inpatient. So, we do it on a daily basis or on a weekly basis. We go through the antimicrobial chart of our patient*". #Clinician 2

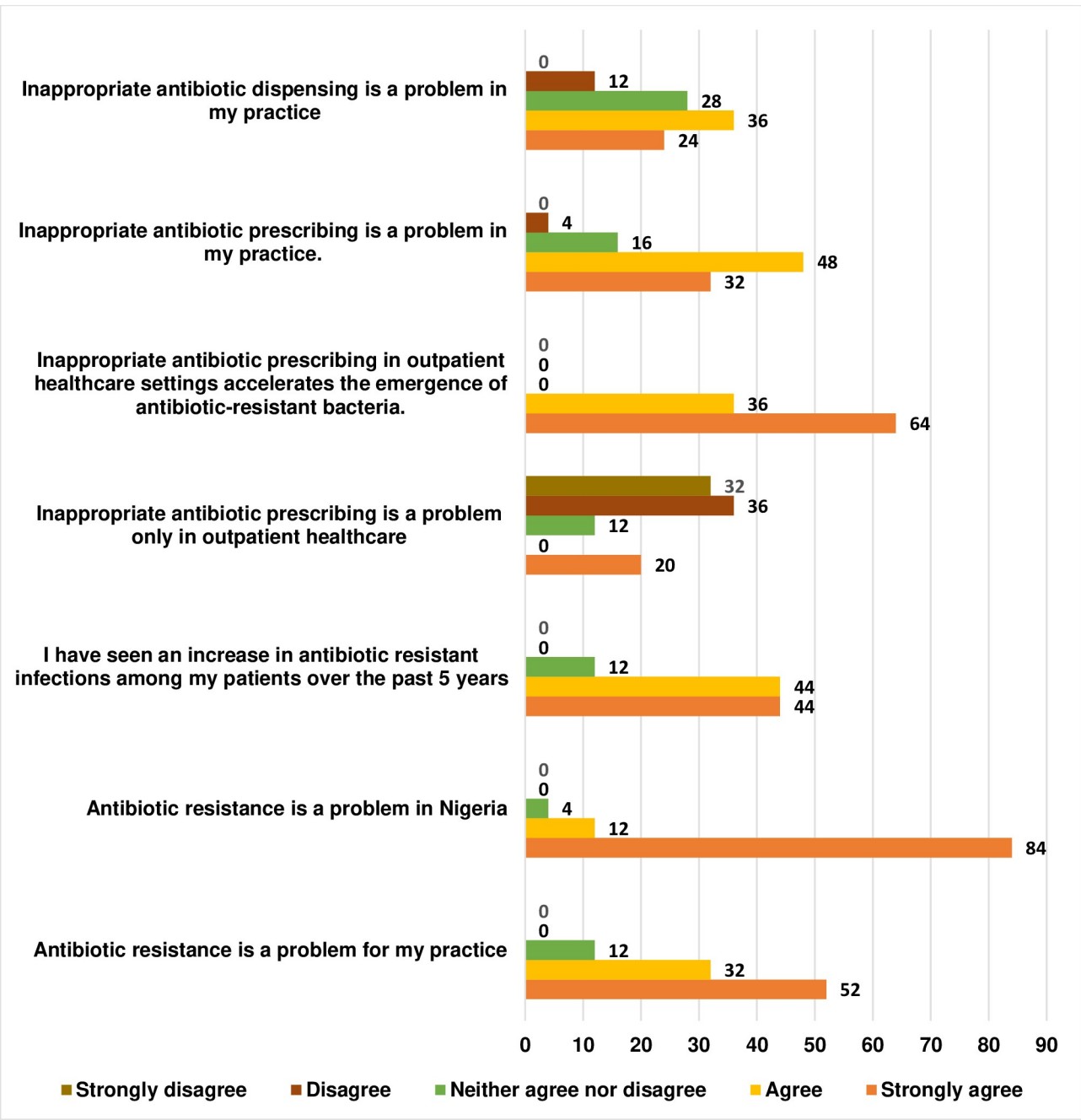

**Fig 3. Responses to questions on the perceptions of stewardship approaches/ interventions.**

However, one of the healthcare facilities had a stewardship program that had been running over a long period of time and were already implementing advanced AMS strategies.

"*We have Antimicrobial Stewardship Committee where we have Clinicians and there are nurses, there are clinical microbiologists from different departments, and we have been running for quite some time which I am a member of the committee. Then so much work has already been done by the stewardship committee. They have developed guideline for some*

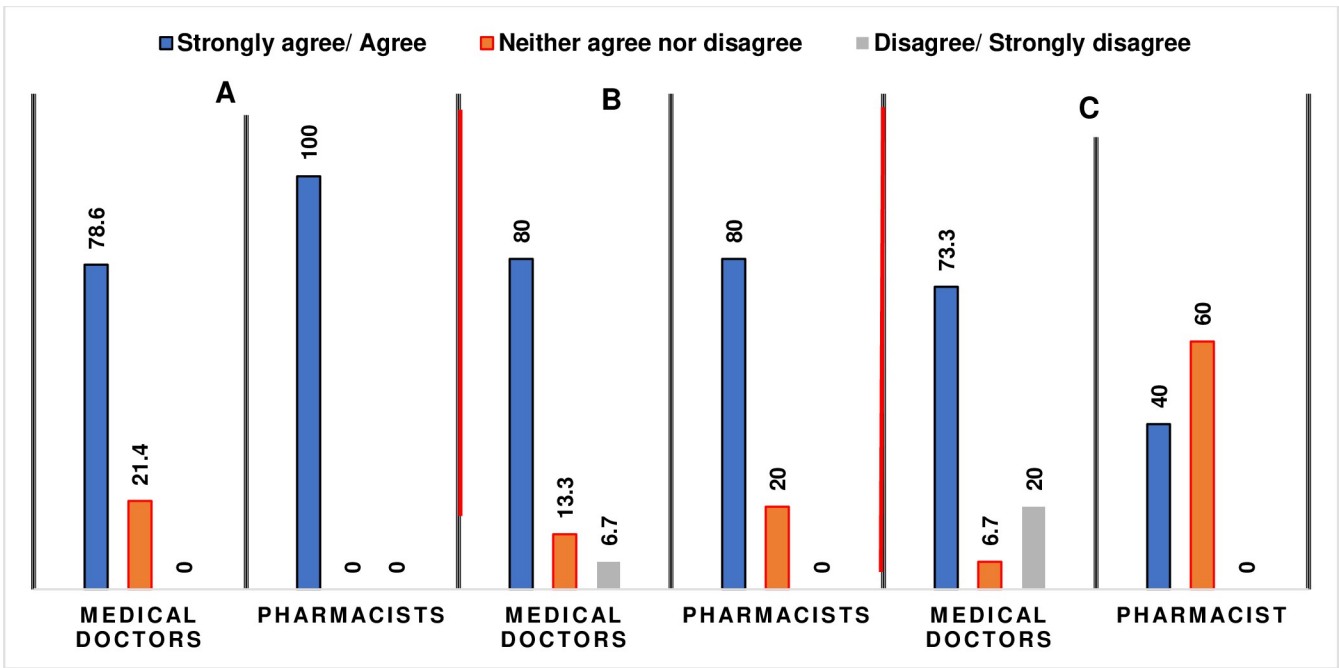

**Fig 4. Comparison of responses of the prescribers to some perception questions.** A = Antibiotic resistance is a problem for my practice. B = Inappropriate antibiotic prescribing is a problem in my practice. C = inappropriate antibiotic dispensing is a problem in my practice.

*departments- medicine. They've also developed antimicrobial use policy and they've done some prevalence surveys in the hospital. They've done some other studies and. . . So, the committee is very. . .very active*". #Clinician 3 (Public tertiary healthcare facility)

**Facilitators to antimicrobial stewardship.** Facilitators discussed by respondents that could promote AMS included creating specialized and dedicated team, having a data repository and effective monitoring.

*Having specialized and dedicated teams.* Some of the participants (n = 15) were of the opinion that ASPs should be created in health facilities and a team constituted with specialized professionals who would be passionate and dedicated to carry out tasks assigned to them in this regard. They believed that establishing teams in health facilities would make ASP easy to access and monitor while curtailing antibiotics misuse from the micro to the macro level.

"*But very importantly, we need to have dedicated personnel that will be facilitating this antibiotic stewardship. If you leave it open, everybody will just want to do his own thing. There will also be issues of. . .oh we are busy; we have other things we are doing . . ..*" #Clinician 4

"*Then also, manpower is indeed useful. . .everybody has said that already. Manpower is very. . .very. . .useful. People are busy, so I think having a dedicated team can help with facilitating antimicrobial stewardship to make it even better.*" #Pharmacist 2

*Data repository.* A few of the respondents (n = 4) were of the opinion that data on AMR should be captured in a local repository. They believed that having data on AMR would paint the true picture of the situation in the country and would help with baseline data for proper assessment as well as effectively implementing strategies to address salient issues related to

antibiotic, its use and/or misuse. "*So, we know that. . . some antibiotics are no longer effective for some infection anymore. But what are the data saying? Like the exact. . .trend in each antibiotic. Making that data available in a repository that is accessible to prescribers will go a long way and also be interesting to look at*". #Clinician 4

"Having accessible data that captures the true situation will give us a better picture to know the right steps to take" #Pharmacist 9

*Monitoring.* Most participants (n = 17) suggested monitoring as a strategy to assess ongoing activities, evaluate the level of progress made or setbacks recorded. Effective monitoring would ensure that tasks are conducted appropriately at the healthcare facility level, which would cascade to other facilities within the state and the country. Furthermore, monitoring will help curb illegal prescription, sanction perpetrators, reduce the abuse of antibiotics and AMR.

"*Antimicrobial stewardship has to be monitored just like we do in pharmacovigilance, and we need to start pushing it about the same way we are doing presently for pharmacovigilance. Such that it should become like a norm. . .it should become compulsory in all institutions*". #Pharmacist 7

"*Then, another aspect is. . . monitoring of the local pharmacy stores. . . . . . . . .particularly those patent medicine stores, the ones in the community. These people. . .they are not trained; they are not pharmacists. . .and they prescribe these antibiotics. So, I think they need to be constantly monitored through a monitoring team to curb illegal prescription*". #Pharmacist 1

**Barriers to the uptake of stewardship programs.** Respondents expressed genuine concern on the implementation of ASPs and might struggle with it due to certain perceived factors. Discussions on the barriers to AMS were summarized under five themes as follows.

*Lack of data on AMR.* Participants (n = 8) believed that there was paucity of data on AMR, which makes it difficult to access the true situation, address appropriately and proffer strategies to ameliorate future incidents from occurring. They proposed that health facilities should set up several databases that would capture activities regarding AMS and AMR with the goal to produce data to highlight stewardship activities and AMR profile, which could help in planning and executing programs. As expressed, "*The truth is that we do not have facility-specific database to show the trend of AMR so it is difficult to make a case.*" # Clinician 10

"*We need data on antibiotic use to inform the hospital management about the prevalence. . .at least if we have data on the pattern of antibiotics consumption at the national, subnational and facility level, then we can start from there to make a case*". #Pharmacist 1

*Inadequate personnel.* Majority of the respondents (n = 17) stated that there were a lot of untrained and unskilled personnel who prescribed antibiotics in the country, and little has been done to stop this. "*We have a lot of unqualified people prescribing, and this is causing more harm in the society.*" # Pharmacist 7. They believed that shortage of staff is also a major limitation to effective stewardship approach and if this trend continued, issues surrounding AMR would not be adequately addressed and this would worsen the situation in the country. "*That's what I mean because most practices are so busy. . .there is shortage of staff. So, nobody wants to take responsibility. If this kind of meetings can be brought closer to the facility with involvement of facility management, then they are more aware and then, maybe. . .can sit down with their antimicrobial stewardship teams and discuss the way forward*". #Pharmacist 10

*Legislation and policies.* Sixteen respondents believed that there were inadequate legislation and policies to guide AMS in the country. They advocated for key stakeholders to be involved in planning, decision-making, and implementation at various spheres at healthcare facilities, pharmacies, state and national level to ensure that the right activities were conducted, and standards were maintained. In addition, policies should be put in place to regulate activities, sanction perpetrators and drive strategies for effective and sustainable implementation and management.

"*There is always so much we can do through education. Education is important, I agree with you. But make laws first. If we don't make laws, we can never get ahead in these things. The truth is that anybody can get any antibiotics they want over the counter, and we are still talking about this. I think we must be stricter with our policies*". #Pharmacist 5

"*I think we should take it up from. . .the government. . . from the top*, *policy formulation, augmentation of those policies. Then down to the. . . . . .end-users of these drugs. The government needs to strengthen existing policies and regulations on the registration of companies involved in drug importation to curb the importation of sub-standard drugs and sanction offending companies*".# Clinician 1

*Lack of focal persons engagement and teamwork.* Participants (n = 16) stated that the managements at the different health facilities, state and federal government have not engaged the right professionals which has had rippling effects and consequences. They believe that the right staff/professionals should be employed or deployed to teams and departments that focused on AMS. They believed that by so doing, the appropriate personnel would have been trained and would possess the requisite skills. Furthermore, it would be easier to orientate and train them based on their educational background and professional expertise. "*The management at the different health facilities need to ensure that they employ staff and constitute a dedicated team that will be trained as focal persons*" # Pharmacist 1. The respondents also identified lack of critical collaboration and teamwork among key stakeholders as a major barrier and maintained that having dedicated staff working in teams will make the program effective.

"*We need to work as a team whether we like it or not. All stakeholders should be represented in the committee in the right proportion. And people that are in the committee should be able to work as a team irrespective of . . .. if you are a pharmacist or a doctor. It should not be a case of . . . . . . why a pharmacist would tell me a consultant what to do, . . .. why should I do this*? *So, that's why I said the willpower will have to be there and everybody should be willing to target the success of the program. . ..*" #Clinician 10

*Extreme multitasking.* Thirteen respondents stated that due to limited personnel who are overburdened with a lot of tasks at the healthcare facilities, it would be difficult to invest their attention on AMS. As expressed, "*In the facility, we are overwhelmed with a lot of work, and it is difficult to combine all the activities effectively. . .it is the same people doing the same job.*" Clinician 10. "*The problem with these committees is that most of these people are doing many other things. And then you want them to get involved with antimicrobial stewardship program*" #Clinician 4. They proposed employing more staff to reduce the burden of work, create comfortable and conducive environments so that more attention can be paid to antimicrobial stewardship.

"*The management should. . .consider employing more qualified personnel to ensure that workload is evenly distributed and. . .. maybe less burdensome to engage in antimicrobial stewardship*" #Pharmacist 3

*Illustrative themes that highlight recommended strategies to improve antimicrobial steward-ship*. Discussions on the recommendations for effective ASPs identified stakeholder's engage-ment, policies, and regulation; and education as themes for improving stewardship in the country.

The identified themes and sub-themes have been summarized in Table 2.

**Table 2. Major themes and sub-themes for improving antimicrobial stewardship and representative quotes.**

| Themes | Description of themes | Sub-themes | Representative quotes |
|---|---|---|---|
| **Stakeholders Engagement** | Fourteen participants from the FGDs advocated for stakeholders' engagement as a strategy for improving AMR stewardship in Nigeria. Stakeholders' engagement was described as involving, advocating and imbibing all key players ranging from policy makers, regulators, healthcare providers, manufacturers, distributors and end-users in appropriate usage, prescribing, selling and consuming of antibiotics. This strategy was recommended because they believed that if stakeholders were educated, awareness will be created. Thus, they would be predisposed to appropriate practices at all healthcare facilities and pharmacies. | **Creating awareness** | *"There is a need to inform and train people about using antibiotics properly so that they can get the knowledge"* #Clinician 2<br>*"I'm talking about schools,.. let's embed antimicrobial stewardship in our curriculums even as early as primary school. At the primary school level, they already teach them about drug abuse, drug misuse. So, this can be expanded upon to teach them about antimicrobial use, misuse and abuse". #Pharmacist 6* |
| | | **Resources/Manpower** | *"It is important to have dedicated team/ personnel that would be mostly involved in antimicrobial stewardship "#Clinician 1*<br>*"Antimicrobial stewardship should be made as a career path for undergraduates and hospitals should employ people whose job is to focus on antibiotics use in the hospital to provide expertise #Pharmacist 10* |
| **Policies and Regulation** | Nineteen participants believed that if rules were set up and guided by strict regulatory bodies, improper use of antibiotics would be significantly reduced in the country. Regulations are rules, processes or directives made and maintained by an authority to oversee activities. Participants proposed that regulatory bodies should be put in place to monitor activities. | **Regulatory roles** | *"Rules and laws should be enacted to guide the distribution and prescription of antibiotics by professionals. Also there should be periodic supervision to health facilities and pharmacies where antibiotics is prescribed "# Pharmacist 7*<br>*"There should be regulatory teams set up to monitor Antimicrobial stewardship activities at different levels in the country"# Clinician 6* |
| | | **Sanctions** | *"There should be punishments for violators which will serve as an example for others to obey "# Clinician 5*<br>*"People who are not trained but are prescribing antibiotics should be identified and sanctioned "# Pharmacist 8* |
| **Education and training** | Nineteen participants believed that education and training was key to achieving AMS and this should involve the key stakeholders and not just a few persons but including doctors, nurses, pharmacists and the management team to ensure an impactful outcome. The participants believed that creating awareness about antibiotics, its use and abuse would help educate, not just prescribers but also facilitate knowledge among end-users thus helping them take the right steps to make proper health decisions. If end-users are educated, they would seek appropriate channels to access health facilities and would take antibiotics only when prescribed by a physician. | **Role Training professionals/ Specialized staff** | *"Clinicians/practitioners need to train themselves and have the right mentality so as to have a good knowledge of antibiotic resistance" # Clinician 4*<br>*"Meetings and trainings concerning Antimicrobial stewardship should be conducted frequently within facilities or made accessible so that it reaches the right professionals "# Pharmacist 3* |
| | | **Educating End-users** | *"Awareness should be created using mass media platforms so that end-users know the dangers of abusing antibiotics "#Pharmacist 3*<br>*"I work in the Internal Medicine department, and I believe educating the patient too is important. . .if we have. . .. probably someone who is trained in antimicrobial stewardship to educate patients at least in the hospital out-patient, we have quite a lot of patients we can find in all these public hospitals. So, by the time we do that, I believe we can virtually educate a lot of people outside there. . .not just the clinicians alone. . . if we can educate a lot of people then we are closer to winning this war. . .at least getting somewhere that we are pushing" #Clinician 4*<br>*"Patients should be taught to go to registered hospitals to access healthcare so that if needed, antibiotics would be prescribed by the specialist "#Clinician 6* |

## Discussion

Antimicrobial stewardship has garnered attention recently as a major means of optimizing antimicrobial use and mitigating AMR. This study identified contextual factors that can mitigate successful implementation of AMS in healthcare facilities in Lagos State. Several hospitals are still facing challenges in the implementation of ASPs despite efforts by CDC in outlining core elements to facilitate implementation of AMS in healthcare setting [12]. Our respondents described various levels of AMS in their facilities with only one facility [tertiary hospital] describing a well-structured and functional stewardship program as recommended by CDC [12]. However, many of the respondents were confident that they can provide input on the implementation of AMS at the facility level although final decision-making still lies in the hands of the facility management. This is consistent with a previous report which revealed minimal ASP activities in healthcare facilities in Lagos State with sub-optimal performance in implementing facilities [14].

Appropriate antimicrobial use has been linked to improved patient outcomes and decreased risk of adverse events, including development of AMR [24]. Although majority of the respondents agreed that ASPs are needed in healthcare settings to effectively deal with AMR, lack of data on AMR was identified as a major obstacle to the uptake of ASP. Availability of cohesive data on AMR has been the bane of most resource poor countries with available data fragmented and lacking in representativeness [25]. Current efforts are being made to collect and collate AMR surveillance data via the GLASS program following the implementation of the Nigerian national action plan for AMR [9, 10]. However, a lot more still needs to be done to decentralize data and include grass root healthcare facilities in the surveillance. Prescribers undoubtedly require data on AMR to be able to fully assess situations and make rational judgements to guide treatment and practice and these data need to be continually updated to reflect current reality. The respondents advocated for facility-specific databases tasked with aggregating AMR data, as a practical approach to solving the problem of dearth of data especially in the Nigerian context.

There were varied perspective among respondents on the problem of AMR with the pharmacists more likely to acknowledge that AMR was a problem for their practice than the medical doctors. This result is consistent with a survey of primary care physicians' attitude and perception towards AMS where only 55% of the physicians agreed that antibiotic resistance was a problem in their practice [19]. This inability to take personal responsibility poses a great challenge to the success of ASP. Some of the clinicians believed that AMR is a problem majorly emanating from unqualified practitioners who are prescribing antibiotics without adequate training. Nevertheless, both professionals jointly agreed that inappropriate prescribing of antibiotics is a major driver of AMR and a problem in their practice (See Fig 4).

Facilitators suggested by participants that could promote AMS included creating specialized and dedicated team, effective monitoring, and establishment of data repository. Some of these factors differ slightly from what has been reported elsewhere [8, 20, 26], indicating contextual undertone. A study on the perspective of pharmacists identified a supportive organizational culture, protected time for antibiotic stewardship, and a cohesive organizational structure as facilitators of effective AMS [27]. These disparities in identified enablers across regions highlights the need for strategic implementation of ASPs with contextual considerations. Majority of the facilities involved in this study are still at the very preliminary stages of establishing ASPs in their facilities hence provide information on perceived facilitators. Nevertheless, the suggested facilitators corresponded to the essential elements identified by different guidelines for setting up and implementing AMS in hospital and out-patient setting in LMIC [12, 24, 28]. The different guidelines and tool kits have their peculiarity but with a common goal of providing strategic steps for implementation of AMS in healthcare settings.

Major themes that emerged as barriers to implementing ASP included inadequate personnel, legislation and policy, engaging the right focal persons and teamwork, multitasking and lack of data on AMR. These contextual issues would necessarily need to be addressed if we are to make meaningly progress as a nation in the antibiotic stewardship effort. A qualitative study to assess the feasibility of the WHO tool kit for AMS in LMIC identified lack of human and financial resources, inadequate regulations for prescription antibiotic sales, and insufficient AMS training as common barriers to implementation of ASP [29]. A few of the respondents indicated that practice-based reporting would be too burdensome and further lamented inadequate personnel as a barrier to proper implementation of stewardship strategies which would require collective and multidisciplinary effort. The respondents opined that in most cases, the available manpower is stretched thin and forced to multitask thereby hindering the likelihood of tracking of appropriate use of antibiotics in an accurate and fair manner. The majority of the respondents believed that the lack of adequate personnel to engage in AMS activities has contributed to several unqualified personnel engaging in antibiotics prescription without adequate training. Similar barriers have been identified by a systematic review of ASPs in developed and developing countries [30, 31]. On the contrary, a multi-country study identified "prohibitively expensive antimicrobials, limited antimicrobial availability, resistance to changing current practices regarding antimicrobial prescribing, and limited diagnostic capabilities" as barriers to improving antimicrobial prescribing [8].

The majority of the respondents agreed that antibiotic resistance is a major problem in Nigeria requiring urgent attention and should be prioritized and positioned at the political level with parliamentary approval. The need for AMS policy embedded in clinical governance and organizational investment in personnel had been identified in a previous study as crucial for a thriving ASP [30]. Nonetheless, successful institution and widespread implementation of ASP in healthcare facilities in Lagos State and Nigeria in general would require prescribers taking individual responsibility for their roles in the emergence or mitigation of AMR. Respondents were very receptive of ASP and willing to engage but they reasoned that implementation of AMS would require committed and dedicated personnel for effective strategic steps.

The respondents identified three themes for improving AMS in the country namely, stakeholder's engagement, enhanced policies and regulation; and education. Majority strongly believed that if policies were set up from the micro level at the healthcare facilities, pharmacies to the macro level at the legislature and Federal Government sphere, illegal distribution, wrong prescriptions, and quack activities will be regulated. Furthermore, they opined that the relevant key stakeholders who had invested in AMS and understood how to navigate the terrain better should be involved in the decision and policy making process for effective outcomes. A study exploring factors influencing implementation of AMS across three low- and middle-income countries [Sri Lanka, Kenya, and Tanzania] recognized 'Improved education and training' as crucial for improving ASPs in tertiary care settings [8]. Education and training remain an integral part of the CDC recommended Core Elements of Hospital Antibiotic Stewardship Program [11]. The respondents reckoned that effective Antibiotics stewardship efforts should be paired with efforts aimed at educating prescribers about antibiotic resistance and antibiotic use. The respondents collectively agreed that holistic approach to awareness creation should target both prescribers and the patients accessing care. Educating the patients about AMR and how inappropriate antibiotic use drives resistance is crucial and will ultimately promote rational use of antibiotics among the public.

This study had some limitations. Study participation was restricted to only the prescribers attending the AMS workshop, hence may not be a true representative of prescribers in Lagos State. However, effort was made to include the different categories of healthcare facilities in Lagos i.e., primary, secondary, tertiary and private/faith-based healthcare facilities. Employing

both quantitative and qualitative method of data collection enabled us to not only obtain information on the availability and level of ASP but also provided a deeper and more holistic understanding of the issues around implementation of ASP. The study did not employ in-depth interviews to gain individual perspective but rather utilized focus group discussions by gathering individual stakeholders for a collaborative discussion on issues around implementing ASPs in their various facilities. Respondents were vocal and understood the thematic focus and provided comprehensive discussions with demonstration of rich understanding of the study context. Majority of the facilities involved in this study (except for one) did not have a structured antimicrobial stewardship program, rather had stewardship teams operating at various levels. However, the respondents provided insight on their perceived facilitators and barriers to implementation of a standard stewardship program in their facilities. Most of the responses were context specific for resource poor setting. Despite using a mixed method, this study did not set out to determine facility specific differences in perceptions towards implementation of ASPs across the different healthcare facilities.

## Conclusion

This study used a mixed method of quantitative and qualitative approach to explore perceptions and experiences, facilitators and barriers to AMS, and recommendations made to key stakeholders with the hope to implement them at various health facilities and pharmacies in Lagos. The study provides evidence-based recommendations to inform future interventional studies focused on implementing ASPs in health facilities throughout Lagos State. The results give insight into the prescribers' perspective on the facilitators and barriers of AMS. Additionally, they shed light on the challenges faced and potential solutions for implementing ASPs in health facilities within Lagos State. Moreover, the study identified crucial contextual factors that should be addressed when developing ASPs in healthcare facilities across the country. These recommendations are even more crucial as the country prepares for the second phase of the National action plan (NAP 2.0) to tackle AMR. These findings provide a solid foundation for developing targeted and effective evidence-based strategies to enhance antimicrobial stewardship practices, ultimately promoting more responsible and sustainable use of antimicrobial agents.

## Supporting information

**S1 Checklist.**
(DOCX)

**S1 Questionnaire.**
(PDF)

**S1 File.**
(PDF)

**S1 Data.**
(XLSX)

## Acknowledgments

The authors wish to acknowledge the following persons: Olanrewaju Ishola, Kelechi Obiozor, Joy Jonah and Fredrick Fadunsin for their secretarial and technical assistance. We appreciate the prescribers who participated in the study and the healthcare facilities in Lagos State that nominated the workshop participants.

## Author Contributions

**Conceptualization:** Emelda E. Chukwu.

**Data curation:** Emelda E. Chukwu, Dennis Abuh, Ifeoma E. Idigbe, Kazeem A. Osuolale, Vivian Chuka-Ebene, Oluwatoyin Awoderu, Rosemary A. Audu, Folasade T. Ogunsola.

**Formal analysis:** Emelda E. Chukwu, Ifeoma E. Idigbe, Kazeem A. Osuolale.

**Investigation:** Emelda E. Chukwu, Dennis Abuh, Ifeoma E. Idigbe, Kazeem A. Osuolale, Vivian Chuka-Ebene, Oluwatoyin Awoderu, Rosemary A. Audu, Folasade T. Ogunsola.

**Methodology:** Emelda E. Chukwu, Dennis Abuh, Ifeoma E. Idigbe, Kazeem A. Osuolale, Vivian Chuka-Ebene, Oluwatoyin Awoderu, Rosemary A. Audu, Folasade T. Ogunsola.

**Project administration:** Emelda E. Chukwu, Dennis Abuh, Ifeoma E. Idigbe, Kazeem A. Osuolale, Vivian Chuka-Ebene, Oluwatoyin Awoderu, Rosemary A. Audu, Folasade T. Ogunsola.

**Resources:** Emelda E. Chukwu, Dennis Abuh, Ifeoma E. Idigbe, Kazeem A. Osuolale, Vivian Chuka-Ebene, Oluwatoyin Awoderu, Rosemary A. Audu, Folasade T. Ogunsola.

**Software:** Emelda E. Chukwu, Kazeem A. Osuolale.

**Supervision:** Emelda E. Chukwu, Rosemary A. Audu, Folasade T. Ogunsola.

**Validation:** Emelda E. Chukwu, Kazeem A. Osuolale.

**Visualization:** Emelda E. Chukwu.

**Writing – original draft:** Emelda E. Chukwu, Ifeoma E. Idigbe.

**Writing – review & editing:** Emelda E. Chukwu, Dennis Abuh, Ifeoma E. Idigbe, Kazeem A. Osuolale, Vivian Chuka-Ebene, Oluwatoyin Awoderu, Rosemary A. Audu, Folasade T. Ogunsola.

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
