## [Decision Letter · Decision Letter 0]

24 Oct 2023

PONE-D-23-27149Implementation of antimicrobial stewardship programs: a study of prescribers’ perspective of facilitators and barriersPLOS ONE

Dear Dr. Chukwu,

Thank you for submitting your manuscript to PLOS ONE. After careful consideration, we feel that it has merit but does not fully meet PLOS ONE’s publication criteria as it currently stands. Therefore, we invite you to submit a revised version of the manuscript that addresses the points raised during the review process.

We look forward to receiving your revised manuscript.

Kind regards,

Mabel Kamweli Aworh, DVM, MPH, PhD. FCVSN

Academic Editor

PLOS ONE

Journal Requirements:

4. Please upload a copy of Supporting Information Figure/Table/etc. Supplementary material (Suppl 1&2) which you refer to in your text on page 4.

Reviewers' comments:

Reviewer's Responses to Questions

**Comments to the Author**

1. Is the manuscript technically sound, and do the data support the conclusions?

Reviewer #1: Partly

Reviewer #2: Partly

2. Has the statistical analysis been performed appropriately and rigorously? 

Reviewer #1: No

Reviewer #2: Yes

3. Have the authors made all data underlying the findings in their manuscript fully available?

Reviewer #1: No

Reviewer #2: No

4. Is the manuscript presented in an intelligible fashion and written in standard English?

Reviewer #1: No

Reviewer #2: Yes

5. Review Comments to the Author

Reviewer #1: 1. General comment: This topic is key in the fight against AMR and the authors are commended for the efforts. However, I strongly suggest that the authors read through the entire manuscript for corrections of some grammatical errors.

2. Abstract

a. In the methods, I suggest you use “focus group discussion” instead of “focused group discussion”.

b. In the methods, no clear details on the analysis of the quantitative data.

c. In the results, you may write “thirty” or “30” instead of “thirty (30)”.

d. Regarding the means, having stated the mean and standard deviation, there may be no need to still state the range. Range is used as the measure of dispersion for skewed data while standard deviation is used as the measure of dispersion for data with normal distribution.

e. The use of AMS in the results was not defined. The authors may have to write it out in full as was done in the conclusion.

f. Having defined ASPs, there should be consistency in its use rather than writing in full as seen in the last sentence of the conclusion.

3. Introduction

a. Define AMR in its first use before continuing with the abbreviation.

b. Having defined ASPs, there should be consistency in its use rather than writing in full again.

c. In page 3, LMICs should be stated in the first paragraph before its continuous use.

4. Methods

a. In the methods, I suggest you use “focus group discussion” instead of “focused group discussion”.

b. Under the “Study Design and Sampling Technique”, some details on the analysis of the qualitative data should be moved to the data analysis subsection of the methods.

c. The authors stated in the results section that 20 participants were interviewed for the qualitative component of the study. Were these key informant interviews or FGDs? If they were FGDs, how many FGDs were conducted and what was the number of participants per FGD? Or were all 20 participants included in one FGD?

5. Results

a. Take note of 2c and 2d above.

b. The results should be reported in past tense. See example from the results section - “About one-half of the participants either do not have or are not sure if there exists an antimicrobial stewardship team in their facility while 28% were members of their facility stewardship team”

6. Discussion

a. The authors made a good attempt to discuss their results.

7. References

a. About half of the references are more than 5 years old. I recommend that the authors add more recent references (within the last 5 years).

Reviewer #2: This work is so important, particularly as Nigeria is implementing NAP. The study will make a great contribution. However, there are issues that need to be considered and are highlighted below. There are valuable questions about the methods and results that need to be addressed to give this readers that insight that is required for this valuable study.

Abstract…… programs and interventions in one sentence side-by-side

…..creating awareness and training and inadequate personnel respectively “not clear”. OBSERVE PUNCTUATION MARKS. Which are the facilitators, and which are the barriers. That needs to clearly come out in your abstract.

……”resource-poor” and not resource poor

Introduction

….”emergence and transmission of AMR in Nigeria” change to “transmission of AMR genes among disease pathogens”. Antimicrobial resistance doesn’t seem to be transmitted. It’s the genes that are the problem.

Change ….”low-and-middle” to ….”low- and middle-income….” That isn’t one word

Methods….

…..”antibiotic- related” change to antibiotic-related

….”faith- based” change to faith-based

What ethical issues could be anticipated from this study and how were these issues addressed?

How many FGDs were conducted and do the authors think the number of FGDs were sufficient?

How about conducting In-depth interviews with these stakeholders? Could that had given better perspective to this study?

What software was used for the qualitative coding and analysis?

……”SPSS version 26.0For…..” Change to ………….SPSS version 26.0. For………….

Results

……”About one-half”……..Correct to “About half” Is there anything like two-half???

68% is about two-third and not about one-half, kindly correct!!!

What the authors described as facilitators are suggested approaches to improve AMS rather than facilitators!! Correct the title to reflect appropriately what is reflected by the results!!

The entire barrier section has not supporting quotes! Why??

What is the difference between the recommended strategies and facilitators as captured in the study??

6. PLOS authors have the option to publish the peer review history of their article (what does this mean?). If published, this will include your full peer review and any attached files.

Reviewer #1: No

Reviewer #2: No

---

## [Author Response · Author response to Decision Letter 0]

9 Nov 2023

Reviewer #1

S/N Reviewer’s comments Authors response

1 General comment: This topic is key in the fight against AMR and the authors are commended for the efforts. However, I strongly suggest that the authors read through the entire manuscript for corrections of some grammatical errors. Thank you for taking time out to review our manuscript. 

The authors have read through the manuscript and corrected grammatical errors 

2 Abstract

a. In the methods, I suggest you use “focus group discussion” instead of “focused group discussion”.

b. In the methods, no clear details on the analysis of the quantitative data.

c. In the results, you may write “thirty” or “30” instead of “thirty (30)”.

d. Regarding the means, having stated the mean and standard deviation, there may be no need to still state the range. Range is used as the measure of dispersion for skewed data while standard deviation is used as the measure of dispersion for data with normal distribution.

e. The use of AMS in the results was not defined. The authors may have to write it out in full as was done in the conclusion.

f. Having defined ASPs, there should be consistency in its use rather than writing in full as seen in the last sentence of the conclusion.

Done!

A statement of the quantitative data analysis approach used has been added.

Corrected

This has been corrected as mean and standard deviation.

This has now been written in full at its first appearance.

Done!

3 3. Introduction

a. Define AMR in its first use before continuing with the abbreviation.

b. Having defined ASPs, there should be consistency in its use rather than writing in full again.

c. In page 3, LMICs should be stated in the first paragraph before its continuous use.

Done

Done

Done

4 4. Methods

a. In the methods, I suggest you use “focus group discussion” instead of “focused group discussion”.

b. Under the “Study Design and Sampling Technique”, some details on the analysis of the qualitative data should be moved to the data analysis subsection of the methods.

c. The authors stated in the results section that 20 participants were interviewed for the qualitative component of the study. Were these key informant interviews or FGDs? If they were FGDs, how many FGDs were conducted and what was the number of participants per FGD? Or 

were all 20 participants included in one FGD?

Corrected

The analysis details have been moved to the data analysis section.

20 participants volunteered and participated in the focus group discussions. Two FDGs were conducted comprising of 10 participants each. This has been clarified

5 5. Results

a. Take note of 2c and 2d above.

b. The results should be reported in past tense. See example from the results section - “About one-half of the participants either do not have or are not sure if there exists an antimicrobial stewardship team in their facility while 28% were members of their facility stewardship team”

Done

Done

6 6. Discussion

a. The authors made a good attempt to discuss their results.

Thank you

7 7. References

a. About half of the references are more than 5 years old. I recommend that the authors add more recent references (within the last 5 years). Some more recent references have been added

Reviewer #2: This work is so important, particularly as Nigeria is implementing NAP. The study will make a great contribution. However, there are issues that need to be considered and are highlighted below. There are valuable questions about the methods and results that need to be addressed to give these readers that insight that is required for this valuable study.

S/N Reviewer’s comments Authors response

1 Abstract…… programs and interventions in one sentence side-by-side

…..creating awareness and training and inadequate personnel respectively “not clear”. OBSERVE PUNCTUATION MARKS. Which are the facilitators, and which are the barriers. That needs to clearly come out in your abstract.

……”resource-poor” and not resource poor

 Done. 

This has been rephrased for clarity.

Done

2 Introduction

….”emergence and transmission of AMR in Nigeria” change to “transmission of AMR genes among disease pathogens”. Antimicrobial resistance doesn’t seem to be transmitted. It’s the genes that are the problem.

Change ….”low-and-middle” to ….”low- and middle-income….” That isn’t one word

Done

Done

3 Methods….

…..”antibiotic- related” change to antibiotic-related

….”faith- based” change to faith-based

What ethical issues could be anticipated from this study and how were these issues addressed?

How many FGDs were conducted and do the authors think the number of FGDs were sufficient?

How about conducting In-depth interviews with these stakeholders? Could that had given better perspective to this study?

What software was used for the qualitative coding and analysis?

……”SPSS version 26.0For…..” Change to ………….SPSS version 26.0. For………….

Done

Done

Ethical clearance for human subject participation to conduct this study was obtained from the NIMR Institutional Review Board. However, participants were informed of their right to either participate or decline to participate in the study and only participants who gave their informed consent by signing the written consent form were included in the study.

Two sets of FGDs were conducted with 10 participants in each set. One set was conducted with medical doctors and another set was conducted with pharmacists. Participation in the FGDs was voluntary. The FGDs had proper representation of participants based on demographics (Gender, age,) type of practice (public, private), range of experiences (early career, mid-career and advanced career clinicians and pharmacists) and the numbers of specialist in each group (10) providing unique and vast knowledge. The FGDs ended when the participants did not provide new or additional data as the same thematic areas were being re-emphasized and re-explained.

This information has been included in the methods section (Focus Group Discussion)

FGDs and In-depth interviews both offer unique advantages for qualitative research. However, for the purpose of this study, we aimed to gather individual stakeholders for a collaborative discussion on issues around implementing ASPs rather than individual perspective. 

In addition, participants were vocal, and understood the thematic focus and provided comprehensive discussions, rich understanding of the study context.

Data analysis was done manually by a team of qualitative research experts and the processes has been explained in the data analysis section. 

Done

4 Results

……”About one-half”……..Correct to “About half” Is there anything like two-half???

68% is about two-third and not about one-half, kindly correct!!!

What the authors described as facilitators are suggested approaches to improve AMS rather than facilitators!! Correct the title to reflect appropriately what is reflected by the results!!

The entire barrier section has not supporting quotes! Why??

What is the difference between the recommended strategies and facilitators as captured in the study?? 

Done

Thank you, the participants responded based on the question asked about factors that could facilitate antimicrobial stewardship. The response documented were the first-hand accounts of what they believed were the factors that will motivate prescribers and healthcare facilities to engage in antimicrobial stewardship. However, we recognize that some participant-identified facilitators also featured/doubled as recommendations hence the implicated themes have now been moved to and discussed under recommendations. 

Supporting quotes have been added to this section.

The respondents provided insights on the facilitators of AMS which are factors that the respondents believed would favour, motivate, or help individual prescribers as well as healthcare facilities to engage in a rational prescription of antibiotics while also recommending approaches for effective ASP implementation.

---

## [Decision Letter · Decision Letter 1]

29 Nov 2023

PONE-D-23-27149R1Implementation of antimicrobial stewardship programs: a study of prescribers’ perspective of facilitators and barriersPLOS ONE

Dear Dr. Chukwu,

Thank you for submitting your manuscript to PLOS ONE. After careful consideration, we feel that it has merit but does not fully meet PLOS ONE’s publication criteria as it currently stands. Therefore, we invite you to submit a revised version of the manuscript that addresses the points raised during the review process.

We look forward to receiving your revised manuscript.

Kind regards,

Mabel Kamweli Aworh, DVM, MPH, PhD. FCVSN

Academic Editor

PLOS ONE

Journal Requirements:

Additional Editor Comments:

In response to the reviewer's comments kindly address the following issues;

1. Please provide additional quotes for facilitators of AMS. For each of the points raised, kindly provide at least two quotes from the FGD. Only one point "Having specialized and dedicated Teams" was supported by one quote.

2. Also provide at least two quotes for each barrier to the uptake of stewardship programs as provided by the respondents during the FGD.  As it currently stands each point has one supporting quote. Kindly provide one additional quote to the one already provided for clarity and consistency please.

3. Once an acronym/abbreviation has been defined upon it's first use, it is expected that you use the acronym in the rest of the manuscript. For instance AMR has been defined previously so kindly correct page 22 line 3 "lack of data on antimicrobial resistance" to "lack of data on AMR". Please fix this issue wherever it occurs in the manuscript. Also check for consistency with other abbreviations used. This applies to AMS and antimicrobial stewardship. So kindly fix this as well all through the manuscript. 

Reviewers' comments:

Reviewer's Responses to Questions

**Comments to the Author**

1. If the authors have adequately addressed your comments raised in a previous round of review and you feel that this manuscript is now acceptable for publication, you may indicate that here to bypass the “Comments to the Author” section, enter your conflict of interest statement in the “Confidential to Editor” section, and submit your "Accept" recommendation.

Reviewer #1: All comments have been addressed

Reviewer #2: All comments have been addressed

2. Is the manuscript technically sound, and do the data support the conclusions?

Reviewer #1: (No Response)

Reviewer #2: Partly

3. Has the statistical analysis been performed appropriately and rigorously? 

Reviewer #1: (No Response)

Reviewer #2: No

4. Have the authors made all data underlying the findings in their manuscript fully available?

Reviewer #1: (No Response)

Reviewer #2: No

5. Is the manuscript presented in an intelligible fashion and written in standard English?

Reviewer #1: (No Response)

Reviewer #2: No

6. Review Comments to the Author

Reviewer #1: (No Response)

Reviewer #2: I still don't understand the reason for leaving out the quotes for some of the facilitation.

The author(s) did not adequately address the issue of barriers. I have stated that the barriers identified by the author(s) did not align with the quotes and read more like suggestions to improvement.

Again, I am wondering if the framework was used to develop a discussion guide because if you begin to ask questions until respondents began to repeat their responses then I wonder of there wad and FGD guide and if it was followed.

Why did the authors decide to leave recommendations in table and the others in full writing?

7. PLOS authors have the option to publish the peer review history of their article (what does this mean?). If published, this will include your full peer review and any attached files.

Reviewer #1: No

Reviewer #2: No

---

## [Author Response · Author response to Decision Letter 1]

22 Dec 2023

S/N Editorial comments Author’s response

1. 

 Please provide additional quotes for facilitators of AMS. For each of the points raised, kindly provide at least two quotes from the FGD. Only one point "Having specialized and dedicated Teams" was supported by one quote.

 Additional quotes have been provided for the facilitators with at least two quotes per listed facilitator 

 Also provide at least two quotes for each barrier to the uptake of stewardship programs as provided by the respondents during the FGD. As it currently stands each point has one supporting quote. Kindly provide one additional quote to the one already provided for clarity and consistency please. One additional quote has been provided for the barriers with at least two quotes per listed barrier

 Once an acronym/abbreviation has been defined upon it's first use, it is expected that you use the acronym in the rest of the manuscript. For instance AMR has been defined previously so kindly correct page 22 line 3 "lack of data on antimicrobial resistance" to "lack of data on AMR". Please fix this issue wherever it occurs in the manuscript. Also check for consistency with other abbreviations used. This applies to AMS and antimicrobial stewardship. So kindly fix this as well all through the manuscript. Done!

The defined terms have been replaced with acronyms all through the manuscript.

 Reviewer #2: . Thank you for your review comments

 I still don't understand the reason for leaving out the quotes for some of the facilitation Quotes have been provided for all the facilitators and barrier initially without quotes and additional quotes added to the existing with a target of two quotes per listed factor 

 The author(s) did not adequately address the issue of barriers. I have stated that the barriers identified by the author(s) did not align with the quotes and read more like suggestions to improvement. The authors presented respondents identified barriers and perceived challenges to the effective implementation of ASP in their various facilities which was summarized under 5 themes. We have added additional quotes to provide more context to their stated barriers.

We also want to mention that majority of the facilities (except one facility) did not have a standard antimicrobial stewardship program, rather had stewardship teams operating at various levels and the respondents discussed their perceived facilitators and barriers to implementation of a stewardship program in their facilities. Most of the responses were context specific for a resource poor setting. This has been stated in the limitation section

 Again, I am wondering if the framework was used to develop a discussion guide because if you begin to ask questions until respondents began to repeat their responses then I wonder of there wad and FGD guide and if it was followed. The FDG was conducted using a semi-structured interview guide which is provided as a supplementary material (S2) with probes to allow flexibility for the respondents provide in-depth insight on emerging themes and sub-themes and also exploring new ideas with follow-up questions until no new information is being provided 

 Why did the authors decide to leave recommendations in table and the others in full writing? The Use of tables for the recommendations allowed for a more detailed presentation of the identified themes and sub-themes and their supporting quotes

However, the implication of all identified facilitators and barriers as well as recommendations were further discussed in context and in comparison, with the quantitative data in the discussion session

---

## [Decision Letter · Decision Letter 2]

8 Jan 2024

Implementation of antimicrobial stewardship programs: a study of prescribers’ perspective of facilitators and barriers

PONE-D-23-27149R2

Dear Dr. Chukwu,

We’re pleased to inform you that your manuscript has been judged scientifically suitable for publication and will be formally accepted for publication once it meets all outstanding technical requirements.

Kind regards,

Mabel Kamweli Aworh, DVM, MPH, PhD. FCVSN

Academic Editor

PLOS ONE

Additional Editor Comments (optional):

Reviewers' comments:

Reviewer's Responses to Questions

**Comments to the Author**

1. If the authors have adequately addressed your comments raised in a previous round of review and you feel that this manuscript is now acceptable for publication, you may indicate that here to bypass the “Comments to the Author” section, enter your conflict of interest statement in the “Confidential to Editor” section, and submit your "Accept" recommendation.

Reviewer #1: All comments have been addressed

Reviewer #2: All comments have been addressed

2. Is the manuscript technically sound, and do the data support the conclusions?

Reviewer #1: (No Response)

Reviewer #2: Yes

3. Has the statistical analysis been performed appropriately and rigorously? 

Reviewer #1: (No Response)

Reviewer #2: Yes

4. Have the authors made all data underlying the findings in their manuscript fully available?

Reviewer #1: (No Response)

Reviewer #2: Yes

5. Is the manuscript presented in an intelligible fashion and written in standard English?

Reviewer #1: (No Response)

Reviewer #2: Yes

6. Review Comments to the Author

Reviewer #1: (No Response)

Reviewer #2: The manuscript looks good but with only one recommendation. "Facilitators" should be changed to "potential facilitators" because they would be facilitators if they are implemented but currently, they are not based on the responses and quotes that are presented at the moment.

The authors need to review the manuscript in accordance with the PLOS ONE guidelines and then it will be good.

7. PLOS authors have the option to publish the peer review history of their article (what does this mean?). If published, this will include your full peer review and any attached files.

Reviewer #1: No

Reviewer #2: No

---

## [Editor Report · Acceptance letter]

11 Jan 2024

PONE-D-23-27149R2 

PLOS ONE

Dear Dr. Chukwu, 

I'm pleased to inform you that your manuscript has been deemed suitable for publication in PLOS ONE. Congratulations! Your manuscript is now being handed over to our production team.

Kind regards, 

on behalf of

Dr. Mabel Kamweli Aworh 

Academic Editor

PLOS ONE